# Genetic Modifiers at the Crossroads of Personalised Medicine for Haemoglobinopathies

**DOI:** 10.3390/jcm8111927

**Published:** 2019-11-09

**Authors:** Coralea Stephanou, Stella Tamana, Anna Minaidou, Panayiota Papasavva, Marina Kleanthous, Petros Kountouris

**Affiliations:** Molecular Genetics Thalassaemia Department, The Cyprus Institute of Neurology and Genetics, Nicosia 2371, Cyprus; coraleas@cing.ac.cy (C.S.); stellat@cing.ac.cy (S.T.); annami@cing.ac.cy (A.M.); panayiotap@cing.ac.cy (P.P.)

**Keywords:** haemoglobinopathies, thalassaemia, sickle cell disease, gene modifiers, biomarkers, gene ranking, protein network

## Abstract

Haemoglobinopathies are common monogenic disorders with diverse clinical manifestations, partly attributed to the influence of modifier genes. Recent years have seen enormous growth in the amount of genetic data, instigating the need for ranking methods to identify candidate genes with strong modifying effects. Here, we present the first evidence-based gene ranking metric (IthaScore) for haemoglobinopathy-specific phenotypes by utilising curated data in the IthaGenes database. IthaScore successfully reflects current knowledge for well-established disease modifiers, while it can be dynamically updated with emerging evidence. Protein–protein interaction (PPI) network analysis and functional enrichment analysis were employed to identify new potential disease modifiers and to evaluate the biological profiles of selected phenotypes. The most relevant gene ontology (GO) and pathway gene annotations for (a) haemoglobin (Hb) F levels/Hb F response to hydroxyurea included urea cycle, arginine metabolism and vascular endothelial growth factor receptor (VEGFR) signalling, (b) response to iron chelators included xenobiotic metabolism and glucuronidation, and (c) stroke included cytokine signalling and inflammatory reactions. Our findings demonstrate the capacity of IthaGenes, together with dynamic gene ranking, to expand knowledge on the genetic and molecular basis of phenotypic variation in haemoglobinopathies and to identify additional candidate genes to potentially inform and improve diagnosis, prognosis and therapeutic management.

## 1. Introduction

Haemoglobinopathies are inherited disorders of haemoglobin (Hb) accounting for over 330,000 annual affected births worldwide. With 5.2% of the global population estimated to carry a potentially pathogenic gene, haemoglobinopathies are the most common monogenic disorders and a serious global public health problem [1]. They are endemic and prevalent in former malaria regions in the Mediterranean, sub-Saharan Africa, the Middle East and South-East Asia, but demographic events, such as global population mobility and migration, have contributed to their spread in all parts of the world [2,3]. As rare disorders in regions with traditionally low incidence and a growing public health burden in resource-limited countries, haemoglobinopathies pose major challenges for health professionals to efficiently diagnose, treat and care for patients [4].

The Hb protein complex comprises two α-like globin chains, encoded by genes in the α-globin locus (Chromosome: 16, Accession: NG_000006), and two β-like globin chains, encoded by genes in the β-globin locus (Chromosome: 11, Accession: NG_000007). The molecular pathology of haemoglobinopathies is traced to genetic defects in the two globin gene clusters, with more than 2000 different mutant alleles reported to date on the IthaGenes database of the ITHANET community portal [5,6]. These mutations can be grouped into those that impair globin chain synthesis, causing thalassaemia syndromes, and those that alter the structure of the Hb protein, causing structural haemoglobinopathies [7]. The pathophysiology and clinical manifestations of haemoglobinopathies are extremely varied with a range of acute and chronic complications that severely impair the quality of life and survival of patients, including iron overload, cardiac siderosis, liver fibrosis, viral hepatitis and endocrine dysfunction for transfusion-dependent thalassaemia, and painful crisis, stroke, acute chest syndrome, pulmonary hypertension, leg ulcers and priapism for sickle cell disease (SCD) [7,8]. Notably, the clinical management and treatment of haemoglobinopathies is challenging as patients with identical genetic defects often present different symptoms, which can even vary in severity over time.

A better understanding of genotype-phenotype correlations and the mechanisms underlying the clinical heterogeneity of haemoglobinopathies not only can improve the management of treatment but can also provide a better chance for the development of personalised medicine. Such knowledge can also enable the identification of affected individuals with a risk for increased disease severity towards early intervention with targeted and preventive care. To this end, β-thalassaemia and SCD, as the commonest of the β-haemoglobinopathies, have been investigated extensively to uncover the genetic determinants in interpatient phenotypic variability. The two best-characterised modifiers are co-inheritance of α-thalassaemia [9,10] and persistence of foetal haemoglobin (Hb F) production [11]. While elevated Hb F levels have no clinical benefit to adults not affected by a haemoglobinopathy, they have been demonstrated to ameliorate disease severity [12,13]. A large number of genome-wide analyses across diverse ethnic populations identified three quantitative trait loci (QTL) modulating Hb F levels: a promoter variant of the Gγ-globin gene (*Xmn*I*-HBG2*), the *HBS1L-MYB* intergenic region (HMIP) and *BCL11A*, which together explain up to 50% of the genetic variation in Hb F [14,15]. Over the past few years, large-scale genome-wide association studies (GWAS) of improved power uncovered additional loci with modest effects on Hb F levels [16,17,18].

Nevertheless, these well-documented modifiers cannot explain the clinical diversity observed among haemoglobinopathy patients. Facilitated by the advent of technology, recent studies have identified variants associated with laboratory and clinical markers of disease severity, such as albuminuria and elevated glomerular filtration rate (GFR) for early renal disease [19], serum lactate dehydrogenase (LDH) for haemolysis [20], abnormal transcranial Doppler velocities for stroke [21] and elevated tricuspid regurgitant jet velocities for cardiopulmonary complications [22,23] (for a comprehensive review see [24]). Measurement of such markers would help risk-stratify patients to direct care, assist with early screening and diagnosis of symptoms, adjust dosing regimens for safe and effective drug therapy, and optimise personalised treatment prior to irreversible tissue damage and organ failure [25,26]. The widespread use of genomic tools provided vast (and still expanding) accumulation of data from association studies with a plethora of publications reporting on significant associations for numerous phenotypes in β-thalassaemia and SCD. 

In the past, data on genetic modifiers of haemoglobinopathies had been scattered across hundreds of published papers, with previous efforts to collect and analyse such data restricted to comprehensive review articles [27,28,29,30] and databases without future updating and annotation [31]. Due to the large volume of literature and the amount of time required to screen and collect relevant data, important information was bound to remain inaccessible to the broad scientific community. Over the past few years, the ITHANET community portal has been curating and annotating disease-modifying genes and variants [5,6], using rigorous literature monitoring. Gene-to-phenotype associations are manually reviewed from the literature by individual assessment and annotated in the IthaGenes database of the portal. With 312 modifier genes and over 600 disease-modifying variants collected from over 450 eligible publications currently annotated in IthaGenes, ITHANET is the first knowledgebase to provide a comprehensive, continuously updated collection of information on genetic modifiers of haemoglobinopathies.

Although such gene-phenotype associations have been freely available on IthaGenes and elsewhere for a few years, the utilisation and analysis of the data has been challenging, owing to the lack of a robust measure to rank available evidence for each gene-phenotype relationship. While experimental validation is an effective approach to deduce strong genetic modifiers from a large number of candidates, it can be laborious and expensive. Alternatively, computational or mathematical methods for gene ranking enable quick assessment of large gene lists to identify top candidates. In fact, several methods have been implemented in the past to evaluate and rank the role of genes in the pathogenicity of different diseases [32,33,34,35,36]. However, similar evidence-based approaches to rank disease-modifying genes have been challenging due to the less prominent role of such genes in disease severity compared to the well-established disease-causing genes and the fact that each modifier gene may influence clinical manifestations for only a small fraction of patients. Moreover, functional analysis of such data and its biological and clinical interpretation have been difficult, and strongly depend on bioinformatics expertise [37].

The present work demonstrates how data organised in IthaGenes can be used by experimental and computational scientists alike to unravel complex gene-phenotype relationships and to explore their relevance in the development of new models of care and therapy for haemoglobinopathies. Specifically, an evidence-based gene ranking algorithm is developed and implemented to study the functional profile of genes that have been linked to modulation of the clinical manifestation and progression of haemoglobinopathies. In addition, functional enrichment analysis, with a focus on protein–protein interaction (PPI) networks as well as pathway and gene ontology (GO) analysis, is utilised to provide insights into the molecular pathology of these diseases and to identify novel target genes for further investigation. Importantly, the analysis revealed functional relationships between curated target genes for selected phenotypes, forming well-connected networks with roles in multiple mechanisms implicated in haemoglobinopathy-specific phenotypes.

## 2. Methods

### 2.1. Data Collection and Preprocessing

The data on disease modifiers were retrieved from the IthaGenes database, which provides a continuously updated, publicly available collection of disease-modifying genes and variations. The content of the IthaGenes database is collected from published peer-reviewed literature using PubMed, through automatic weekly searches for haemoglobinopathy-specific keywords, previously described in Kountouris et al. (2014) [5]. In brief, the titles and abstracts of retrieved publications are screened manually by the IthaGenes Curation Team and, if relevant, the full text is thoroughly examined to extract information on the relationships between genes, variants and phenotypes. The references from each publication are manually filtered to expand information on previously reported gene–phenotype relationships and to identify new disease-modifying variants and genes. Consequently, the final list of articles utilised in IthaGenes describes studies aiming to unravel genotype-phenotype relationships relevant to haemoglobinopathies and include GWAS, linkage, candidate gene, case-level and functional studies. Statistically significant associations (*p* value <0.05) or experimental evidence are then extracted from the articles and used for gene and/or variant annotation in IthaGenes. Each genetic modifier is linked to at least one phenotypic term mapped with standardised annotations curated by the human phenotype ontology (HPO) [38,39]. Those with poor phenotype definitions or terms not contained in HPO are annotated by terms that best describe the clinical characteristics of the study population or laboratory risk factors investigated. Moreover, genetic modifiers are linked to data from existing public databases (e.g., National Centre for Biotechnology Information (NCBI) Gene, Online Mendelian Inheritance in Man (OMIM), Universal Protein Resource (UniProt), Single Nucleotide Polymorphism Database (dbSNP)) and receive a multitude of additional manual annotations, such as gene function, the role in disease and the effect on phenotypes.

As part of this study, the dataset retrieved from IthaGenes was further processed to identify studies that report on the same piece of evidence, e.g., reviews reporting associations described in original studies that had already been included in the dataset. Such duplicated evidence was removed from the dataset, whereas the quality of each study was also further annotated and evaluated by collecting information about the type and design of study, reported *p* values and confidence intervals and use of multiple testing, if needed. The final dataset used for the current study comprises 493 unique gene-phenotype relationships, derived from a total of 312 genes and 59 phenotypes, with data on β-thalassaemia and SCD analysed together as pooled data for β-haemoglobinopathies.

### 2.2. Development of an Evidence-Based Approach for Gene Ranking

The volume of available evidence for each gene–phenotype relationship in the dataset is represented quantitatively with three different scores, namely Association Score, Variant Score and Experimental Score, using a point system to reflect the strength of each piece of evidence. Similar approaches have been developed in the past to quantify existing evidence for gene-disease relationships [36,40,41], but, to our knowledge, this is the first effort to develop an evidence-based framework for modifier genes in a Mendelian disorder. The point system used for each individual score is shown in Table 1 and briefly described below.

The Association Score (AS) represents the sum of points derived from statistically significant associations for each gene-phenotype relationship. For every study in the dataset, the most significant variant of each gene for a given phenotype was selected to represent the strength of the gene-phenotype relationship. Three different evidence levels were considered to score each study for a given phenotype as follows: (a) case-level studies and association studies reporting statistically significant associations with a *p* value of <0.05, scored with 0.5 point, (b) association studies with at least one variant with a *p* value of <0.001, scored with 1 point, and (c) association studies with at least one variant with a *p* value of <10^−5^, scored with 1.5 points. To avoid possible bias from multiple case-level studies (under the lowest evidence level above), a maximum of four case-level studies (i.e., a total of 2 points, with 0.5 point awarded for each case study) were considered for each gene–phenotype relationship. In addition, all association studies were evaluated qualitatively to detect studies with weak methodology (e.g., lacking multiple comparison procedures and confidence intervals). Such cases remained in the dataset to avoid reduction of the evidence pool, but their AS was reduced by a penalty of 25%. Subsequently, the sum of all points from different independent studies was calculated for each gene–phenotype relationship.

The Variant Score (VS) represents the number of variants identified in each gene–phenotype pair and are curated in IthaGenes, the largest database of modifiers relevant to haemoglobinopathies. In each gene-phenotype relationship, a single point was awarded for every variant in the database.

The Experimental Score (ES) represents the sum of all points derived from experimental evidence available for each gene–phenotype relationship. Given that the implication of modifier genes in the pathology of haemoglobinopathies needs to be validated by experiments that support a role for that gene with respect to the phenotype under study, a point system similar to the work of Strande et al. [40] was employed to divide experimental evidence into three main categories: gene function (biochemical function, protein interaction and expression), functional alteration, and model systems (model organisms and phenotypic rescue). Experimental studies on gene function, functional alteration and model systems received 1, 1.5, and 2 points, respectively. The sum of all points derived from experimental evidence was subsequently calculated for each gene–phenotype relationship.

A maximum allowed sum of points was set for each individual score in order to count for multiple replication studies establishing a gene–phenotype, but, at the same time, to avoid overrepresentation (i.e., very high scores) of well-established disease modifier genes in our analysis, such as *BCL11A* and *KLF1*. The maximum allowed scores for AS, VS and ES were set to 8, 20 and 6, respectively. All individual scores for each gene-phenotype pair were subsequently normalised to be canonical (from 0 to 1) by dividing the total score by the maximum allowed sum of points.

The overall score, called IthaScore*,* is calculated with the formula below, using a weighted sum of all individual scores and reflects the available evidence for each gene-phenotype relationship. The weights have been selected to represent both the strength of each evidence type, but also the volume of available evidence in the dataset. Therefore, a stronger weight is used for association studies that represent the overwhelming fraction of evidence in the dataset, with around 85% of scores derived from association studies and 15% from functional studies.

IthaScore=0.5∗AS+0.2∗VS+0.3∗ES

### 2.3. Functional Enrichment Analysis

Functional enrichment analysis was performed for each phenotype in the dataset using their corresponding gene lists. Specifically, the Search Tool for the Retrieval of Interacting Genes/Proteins (STRING) database v.11.0 [42] was used for the construction of PPI networks, followed by GO term enrichment and pathway analysis (Kyoto Encyclopedia of Genes and Genomes (KEGG) and Reactome). Only functional enrichment terms with a *p* value of 10^–5^ or lower (after false discovery rate (FDR) correction), as provided by STRING, were considered for further investigation. The Human Genome Organisation (HUGO) Gene Nomenclature Committee (HGNC) approved gene symbols were used as input data, thereby excluding intergenic regions from functional enrichment and network analysis. Connections (edges) between proteins (nodes) were predicted at a high confidence cut off of ≥0.7 using all types of evidence available in STRING, while the top five additional interactors with the initial gene set were also included in the analysis. High-resolution bitmaps of the PPI networks were displayed and exported from STRING. In addition, GOnet [43] was used to investigate and visualise relationships between specific gene lists and statistically significant GO terms. The Comparative Toxicogenomics Database (CTD) MyVenn tool [44] was utilised to identify common genes between different phenotype-specific gene list in the dataset, specifically “Hb F levels”, “F-cell numbers” and “Hb F response to hydroxyurea”. The Cytoscape software, version 3.7.2 [45], was used to visualise gene-phenotype relationships in a network format.

## 3. Results and Discussion

### 3.1. Exploratory Analysis of Modifier Gene Lists

The dataset was analysed to identify genes and genomic locations involved in several phenotypes relevant to haemoglobinopathies, visualised in Figure 1. Although, as expected, the majority of genes (219 out of 312 genes) were assigned to a single phenotypic term, numerous genes were linked to multiple phenotypes. Notably, eleven genes are assigned to five or more phenotypic terms, of which HMIP, *BCL11A* and *NOS3* ranked top with 10, 11 and 13 phenotypic terms, respectively. Such multiple assignments are expected due to common pathophysiological mechanisms in many disease complications (e.g., haemolysis and vaso-occlusion), thus involving similar sets of genes. The full list of genes ordered by the number of phenotypic annotations is shown in Appendix A.

Figure 2 shows a summary of the 59 phenotypic terms in the dataset ordered by the number of genes and variant annotations. The number of genes and variants differed among phenotypes, with “Hb F levels” (82 genes, 299 variants) being the most prevalent term, followed by “Hb F response to hydroxyurea” (42 genes, 69 variants). Other frequently assigned terms involved clinical descriptions relevant to haemolysis and vaso-occlusion, including: ”stroke”, “osteonecrosis/avascular necrosis”, “pulmonary arterial hypertension”, “pain”, “acute chest syndrome” and ”leg ulcers”. In addition, 36 phenotypic terms were assigned to five or fewer genes, of which thirteen were annotated with a single gene bearing a few variants.

### 3.2. Evidence-Based Gene Ranking

The gene ranking analysis integrated a scoring metric, called IthaScore, where each gene-phenotype interaction was assigned a combined score of various evidence measures (see Methods for details on the calculation of IthaScore). Overall, 483 gene-phenotype interactions were identified and scored, with IthaScore ranging from 0.023 to 0.875. The entire gene list as well as their scores and ranking are shown in Appendix A. Higher gene scores indicate a greater likelihood that genes would have an effect on the phenotypes investigated. Figure 3 shows the distribution of IthaScore for all gene-phenotype relationships, while Table 2 shows the top scoring genes for each of the 59 phenotypic terms in the dataset. Importantly, most of the gene-phenotype pairs have a low IthaScore, thus highlighting that available evidence for those relationships is currently weak and that additional studies are needed before such associations are considered reliable.

In contrast, our approach is validated by its ability to produce a high IthaScore for well-established disease modifiers, particularly those involved in Hb F production and modulation, such as *BLC11A*, HMIP and *KLF1*. This is demonstrated in Table 3, which lists the ten gene-phenotype pairs with the highest IthaScore, with nine of them involved in Hb F modulation. In addition, our method successfully highlights, with a high IthaScore, the well-established role of *UGT1A1* in bilirubin metabolism, especially since genetic variations in *UGT1A1* constitute major risk factors for unconjugated hyperbilirubinemia [47].

A gene-phenotype network was constructed and shown in Figure 4 depicting gene-phenotype relationships with an IthaScore of at least 0.1 to show significant relationships and, also, allow clearer interpretation and visualisation. The edge weights represent the strength of the relationship based on the calculated IthaScore, while each phenotype is labelled with a unique identifier, as defined in Table 2, for better visualisation of the network.

Naturally, phenotype “Hb F levels” (node 2 in Figure 4, Panel A) is a clear hub in the network showing both the highest number of connected genes (with IthaScore ≥0.1) and the strongest connections, i.e., the highest IthaScore in the network. High scoring loci in the “Hb F levels” phenotype, such as *BCL11A* and HMIP, have also weaker connections with other phenotypes in the network, although this can be an indirect effect in the disease severity due to the well-established role of high Hb F as a disease-modifying factor.

As the level of Hb F is a major predictor of survival in haemoglobinopathies, genetic markers that modulate Hb F production have been investigated extensively. Similar to numerous studies reported to date [16,48,49], the top-ranked genes for interaction with “Hb F levels” include *BCL11A* (0.875), HMIP (0.825), *KLF1* (0.711) and *HBG2* (0.6). Another sensitive biological indicator of Hb F is the abundance of Hb F-containing erythrocytes (F cells) [11]. In this analysis and in line to published work [50], *BCL11A*, HMIP, *KLF1* and *HBG2* are ranked as the leading modifiers of “F cell numbers” (node 10). Moreover, hydroxyurea (HU), as a potent pharmacological inducer of Hb F, is used in the treatment of SCD, although with highly variable degrees of clinical response [51]. The search for genetic modifiers of “Hb F response to HU” (node 9) identified associations to *BCL11A* and *HBG2* as the most robust [52]. Although both genes drew top scores following ranking, less prominent Hb F-promoting loci, including *SAR1A*, *MAP3K5*, *NOS1* and *ARG2*, emerged as promising predictors of drug response based on the calculated IthaScore. While many of the Hb F-promoting loci are also associated with Hb F response to HU, the absence of strong Hb F modulators, such as *KLF1* and HMIP, from loci associated with Hb F response to HU suggests that some mechanisms of HU-induced Hb F may differ from mechanisms of endogenous Hb F regulation (candidate mechanisms of HU-induced Hb F are summarised in Pule et al. [53]).

Other smaller subnetworks shown in Figure 4 highlight the role of different genes in other disease phenotypes, specifically including (a) anaemia (node 4), ineffective erythropoiesis (node 5) and abnormal red blood cell (RBC) count (node 1), (b) bilirubin metabolism (node 28) and gallstone formation (node 29), and (c) phenotypes/complications related to vaso-occlusion and/or haemolysis like acute chest syndrome (ACS) (node 14) and stroke (node 3). The above phenotype groups are highlighted in Panels B, C and D of Figure 4 respectively, and discussed below.

Ineffective erythropoiesis is a hallmark of β-thalassaemia characterised by excess free alpha haemoglobin (α-Hb) pool in erythroid precursors, which leads to their premature destruction within the bone marrow, resulting in abnormal counts of RBCs in circulation and, thus, to anaemia [54]. Supported predominantly by functional evidence, *SOX6* and *AHSP* were identified as the leading modifiers of “ineffective erythropoiesis” (node 5), while *AHSP* also achieved high IthaScore for interaction with “anaemia” (node 4). In fact, *AHSP* is a candidate molecular chaperone for free α-Hb and a critical modulator of β-thalassaemia [55]. Additionally, *CCND3* had a high IthaScore for interaction with “anaemia” and “abnormal RBC counts” (node 1), which is in line with its role in controlling cell cycle progression and differentiation during haematopoiesis and thereby RBC size and count [56].

One of the best-known genetic modifiers of bilirubin metabolism and cholelithiasis in haemoglobinopathies is the *UGT1A* locus [27]. As expected, and illustrated in Panel C of Figure 4, members of the UGT1A family, namely *UGT1A10*, *UGT1A6* and *UGT1A1*, were among the top-ranked genes for interactions with “bilirubin levels” (node 28) and “gallstone” formation (node 29). 

Haemolysis and vaso-occlusive phenomena are fundamental features of SCD affecting a variety of tissues and organs [57]. Here, we present candidate genes that could potentially influence two of the most important complications of SCD: ACS (node 14) and stroke (node 3) (Panel D of Figure 4). ACS is a vaso-occlusive crisis of the pulmonary vasculature and one of the leading causes of hospitalisation among SCD patients [58] and has been associated with effects of endothelial nitric oxide (eNOS) metabolism, inflammation, cell adhesion, hypoxia and endothelial damage [59]. As expected, high-scoring genes for “acute chest syndrome” (node 14) included *EDN1* and *NOS3*, as well as genes involved in the TGF-β signalling pathway, namely *TGFBR3*, *SMAD1* and *SMAD7*. Although stroke is one of the most disabling complications, the factors that lead to stroke remain elusive [60]. The top-scoring genes for “stroke” (node 3) included *ENPP1, TGFBR3, ADCY9, BCL11A* and *BMP6*.

### 3.3. Functional Enrichment Analysis for Selected Phenotypes

Towards understanding the biological meaning behind large lists of genes for specific phenotypes and in search for their mechanisms of action, functional enrichment analysis focused on identification of enriched GO terms, specifically biological process (BP) and molecular function (MF), as well as associated pathways (from KEGG and Reactome). Only enriched GO terms and biological pathways with an FDR <10^−5^ were considered. Those associated with a low gene count in the database were more specific, thus giving a greater biological meaning. Given that a complete functional enrichment analysis for each of the 59 phenotypes is beyond the scope of this work, we demonstrate the results of the analysis for three selected phenotypes related to different pathophysiological mechanisms and of different gene set sizes: (a) Hb F levels in relation with Hb F response to HU, (b) response to iron chelators and (c) stroke.

#### 3.3.1. Hb F Levels and Hb F Response to Hydroxyurea

The discovery of genetic markers for the upregulation of Hb F in patients with β-thalassaemia and SCD has been a major ongoing research effort for decades, resulting in a large volume of data in the literature. Drawing information from studies showing a positive correlation between Hb F levels and the number of F cells [61], the gene sets of these two phenotypes were pooled for simplicity (from here on referred to as phenotype “Hb F levels/F-cells”). Additionally, the major benefit of hydroxyurea (HU) on disease severity is directly related to its effect on Hb F production [62]. The large number of reported genes made it challenging to establish informative GO term and pathway rankings with relevance to the “Hb F levels/F-cells” phenotype, instigating the need for further gene set enrichment analysis. As to remove noisy information from the analysis and to identify candidate genes that regulate fetal γ-globin genes and also modulate HU-induced Hb F levels, genes that were common between phenotypes “Hb F levels/F-cells” and “Hb F response to HU” were identified (11 genes) and used as input data for analysis. These included ARG2, ASS1, BCL11A, FLT1, HBE1, HBG2, MAP3K5, NOS1, SAR1A, TOX and VEGFA. Five additional interactors were allowed in the network to identify the most significant interactions to the initial protein list and achieve a meaningful size for network analysis (16 nodes total), shown in Figure 5A. Interestingly, these interactors contained five additional proteins without prior connotation to the above Hb F-related phenotypes, except for the VEGF receptor KDR (kinase insert domain receptor). These new candidate proteins included ASL (argininosuccinate lyase), OTC (ornithine carbamoyltransferase), PGF (placental growth factor) and VEGFB (vascular endothelial growth factor B). In addition, three of the proteins (MAP3K5, SAR1A and TOX) were not engaged in any interactions with the high confidence interaction score 0.7 in STRING. 

The PPI network and the subsequent functional enrichment analysis of the final protein list resulted in two distinct clusters (Figure 5A). One cluster included five proteins, namely ARG2, ASL, ASS1, NOS1 and OTC, that are annotated with GO terms and pathways involved in nitrogen metabolism, including GO terms “urea cycle” (GO:0000050), “urea metabolic process” (GO:0019627) and “arginine metabolic process” (GO:0006525) and pathways “urea cycle” (HSA-70635) and “arginine biosynthesis” (hsa00220). The second cluster contained five proteins, namely FLT1, KDR, PGF, VEGFA and VEGFB, that are linked to functional terms related to the VEGF-VEGFR system, including “positive regulation of angiogenesis” (GO:0045766), “vascular endothelial growth factor receptor signalling pathway” (GO:0048010), and pathways involved in vascular endothelial growth factor (VEGF) ligand-receptor interactions (VEGF binds to VEGFR leading to receptor dimerisation “HSA-195399” and MAPK signalling pathway “hsa04010”). Overall, significant GO terms and pathways were consistent between them, with Figure 5B illustrating interactions between genes and GO term enrichment analysis.

Notably, three of the query proteins (ARG2, ASS1 and NOS1) and two of the new interactors (ASL and OTC) are involved in the urea cycle and the L-arginine biosynthesis sub-pathway (Figure 6). Specifically, argininosuccinase (ASL) catalyses the production of arginine from arginosuccinate, while ornithine carbamoyltransferase (OTC) catalyses the production of citrulline, an intermediate substrate in the pathway of arginine synthesis. Drawing information from studies investigating the factors that are implicated in a variable Hb F response to HU treatment, there is strong evidence to suggest that the arginine-dependent nitric oxide (NO) pathway is involved in the induction of Hb F [63,64,65]. NO is a signalling agent produced from the metabolism of L-arginine by the enzyme nitric oxide synthase (NOS) [66]. The underlying mechanism involves NO-mediated activation of soluble guanylate cyclase (sGC) and subsequent signalling via the sGC/cyclic guanosine monophosphate (cGMP)-dependent protein kinase (PKG) pathway [67]. Considering that this effect can also be mediated by other NO donor substrates, it is important to explore ASL and OTC as potential mechanisms by which drug-mediated NO production could be therapeutic or prognostic of drug efficacy.

Also associated with Hb F levels and Hb F response to HU were proteins involved in vasculogenesis and angiogenesis, namely VEGFA (vascular endothelial growth factor A), FLT1 (vascular endothelial growth factor receptor 1, VEGFR1) and the new interactors VEGFB (vascular endothelial growth factor B) and PGF (placenta growth factor). The mechanism by which these genes influence Hb F production is still unclear, yet a growing amount of evidence implicates an effect on the process of erythropoiesis [68,69,70,71]. Notably, additional studies will be necessary to identify the functional role of VEGF signalling and other potent factors on erythropoiesis, as well as their effect on globin gene transcription programs.

#### 3.3.2. Response to Iron Chelators

Deferiprone and deferasirox are standard drugs for iron chelation therapy in transfusion-dependent anaemias. Decreasing excess accumulation of iron through the use of chelation reduces damage to critical organs [72]. However, patients show different rates of adherence and drug-related toxicities, indicating that genetic factors may influence the way drugs are metabolised [73,74]. To identify potential molecular pathways related to response to chelation therapy with deferiprone and deferasirox, genes associated with phenotypes “response to deferiprone” and “response to deferasirox” were pooled (*ABCC2*, *CYP1A1*, *CYP1A2* and *UGT1A6*) and used as input for the STRING database. Figure 7A shows the PPI network, including five additional interactors, namely UGT1A3, UGT1A4, UGT1A8, UGT1A9 and AHR, and Figure 7B illustrates the interactions between genes and GO term enrichment analysis.

The enriched GO BP terms indicate gene functions mainly associated with (a) xenobiotic metabolism, including “cellular response to xenobiotic stimulus” (GO:0071466), “xenobiotic metabolic process” (GO:0006805) and “flavonoid metabolic process” (GO:0009812), (b) glucuronidation, such as “negative regulation of cellular glucuronidation” (GO:2001030), “negative regulation of glucuronosyltransferase activity” (GO:1904224) and “xenobiotic glucuronidation” (GO:0052697), and (c) fatty acid metabolism, including “monocarboxylic acid metabolic process” (GO:0032787), “negative regulation of fatty acid metabolic process” (GO:0045922), and “omega-hydroxylase P450 pathway” (GO:0097267). Pathways involved in “retinol metabolism” (hsa00830), “metabolism of xenobiotics by cytochrome P450” (hsa00980) and “glucuronidation” (HSA-156588) were also deemed enriched. Notably, only the *UGT1* locus is associated with the glucuronidation pathway.

These findings are in line with published work demonstrating that deferiprone and deferasirox are mainly metabolised by glucuronidation [73,75], a major pathway of xenobiotic biotransformation (phase II metabolism, conjugation) catalysed by uridine 5’-diphospho-glucuronyltransferases (UGT). Members of the UGT1 family are the most important in terms of drug metabolism and are found primarily in the liver [76]. Cytochrome P450 (CYP) is another family of xenobiotic-metabolising enzymes (phase I metabolism, functionalisation) [77], of which only two members (CYP1A1 and CYP1A2) appear in the PPI network. Both CYP1 proteins interact with the aryl hydrocarbon receptor (AHR), a xenobiotic receptor that regulates the activation of CYP1A1, CYP1A2 and several other genes, including UGT1A4, UGT1A6 and UGT1A9 [76,78,79].

Glucuronidation of deferasirox is mainly mediated by UGT1A1 and to a lesser extent by UGT1A3, with minor contributions from UGT1A7 and UGT1A9, and trace activities by several other UGTs (UGT1A4, UGT1A6, UGT1A8, UGT1A10, UGT2B4, UGT2B7, UGT2B15 and UGT2B17) [80]. Oxidative metabolism by CYP enzymes (CYP1A1, CYP1A2 and CYP2D6) has a minor contribution to the elimination process [81]. Deferasirox and its glucuronide metabolites are eliminated mainly by hepatobiliary transport via multidrug-resistance protein 2 (MRP2) [82]. MRP2, also known as ABCC2, is an anion transporter expressed at important pharmacological barriers, such as the canalicular membrane of hepatocytes, with an important role in the elimination of xenobiotic substrates [83]. Moreover, glucuronidation of deferiprone is catalysed almost exclusively by the UGT1A6 in hepatic tissues with subsequent excretion in the urine. Several other UGTs (UGT1A7, UGT1A8, UGT1A9, UGT1A10, UGT2B7, and UGT2B15) exhibit trace activities and are not expected to impact the formation of glucuronide metabolites [75,84,85].

The results of the functional enrichment analysis indicate that the proteins involved in the metabolism and transport of deferiprone and deferasirox may also influence response to therapy. Specifically, new candidate modifiers include UGT1A4, UGT1A8 and UGT1A9, which exhibited low metabolic clearance of these drugs with in vitro animal tissue models. As drug metabolism and interactions are species-specific [86] and given that drug-metabolising enzymes have different rates of maturation at different developmental stages [87,88], further studies are needed to unravel their role in the biotransformation of iron chelators as to better serve patients. Overall, our analysis revealed new genes as candidate pharmacogenetic biomarkers of deferiprone and deferasirox efficacy that seek further investigation.

#### 3.3.3. Stroke

Stroke is one of the most devastating complications of SCD affecting up to 11% of patients with sickle cell anaemia (Hb SS) and sickle β^0^-thalassaemia under 18 years of age without intervention [89,90,91]. Sibship studies demonstrated that stroke has an inherited component and is, therefore, genetically modifiable [92]. However, stroke is a complex process with variability in lesion size, location and etiology, and, thus, unlikely to be modified by a single gene [60]. Genetic susceptibility appears to be guided by many genes with small effect sizes [93]. The dataset consisted of 28 modifiers with diverse functions, including inflammation (*TNF, TGFBR3, IL4R, BMP6, CCL2, LTC4S* and *IL6*), adhesion (*VCAM1, TEK, SELP, CSF2, LDLR* and *ECE1*), coagulation (*ANXA2* and *F5*), signal transduction (*ADYC9, ADRB2* and *AGT*), cell survival (*MET*), oxidative stress (*HMOX1* and *PON1*) and transcriptional regulation (*ERG, HDAC9* and *BCL11A*) [21]. The high genetic heterogeneity reflects the complexity of stroke pathogenicity.

The enrichment analysis of the GO terms and biological pathways was carried out on 33 proteins (Figure 8A), of which IL4 (interleukin 4), IKBKG (inhibitor of nuclear factor kappa B kinase regulatory subunit gamma), RIPK1 (receptor interacting serine/threonine kinase 1), TRADD (TNFRSF1A associated via death domain) and TRAF2 (TNF receptor associated factor 2) were new interactors at the confidence interaction score ≥0.7 (Figure 8B). Interestingly, IKBKG, RIPK1, TRADD and TRAF2 formed a discreet and distinct cluster that linked to the rest of the network via interaction with TNF (tumour necrosis factor). TNF is a pro-inflammatory cytokine produced by brain cells with presence in all stages of brain injury by stroke. It plays a central role during cerebral ischemia and exerts both damaging and protective functions via interaction with the TNF receptor superfamily member 1A (TNFRSF1A, also known as TNFR1). The DD domain of the TNFR1 binds TRADD, which in turn recruits TRAF2, RIPK1 and Fas-associated via death domain (FADD). Binding of TRAF2 with cellular inhibitor of apoptosis proteins (cIAPs) facilitates NF-κB activation and induction of NF-κB-regulated anti-apoptotic factors. The protein IKBKG forms part of the IκB kinase (IKK) complex involved in the activation of NF-κB. On the other hand, activation of RIPK1 and FADD-interacting initiator caspase [FADD-like interleukin-1β-converting enzyme (FLICE)/caspase-8] lead to necrotic or apoptotic cell death (for review see [94,95]). Other potent pro-inflammatory cytokines with a significant impact on stroke pathology include interleukin 1 (IL-1), IL-4, IL-6, IL-8, IL-10 and IL-17 [96,97]. 

Functional enrichment analysis revealed 146 significant GO BP terms (*p* value <10^–5^), of which the top five were annotations to general terms in GO hierarchy (e.g., “positive regulation of multicellular organismal process”—GO:0051240, “response to organic substance”—GO:0010033 and “response to oxygen-containing compound”—GO:1901700). Enrichment analysis of GO MF terms yielded seven significant terms, of which the more relevant (in order of increasing FDR) were “cytokine receptor binding” (GO:0005126), “signalling receptor binding” (GO:0005102), “cytokine activity” (GO:0005125) and “tumour necrosis factor receptor superfamily binding” (GO:0032813). The most prominent pathways involved “TNF signalling pathway” (hsa04668), “IL-17 signalling pathway” (hsa04657), “NF-kappa B signalling pathway” (hsa04064), “IL-4 and IL-13 signalling” (HSA-6785807) and “TNFR1-induced NFkappaB signalling pathway” (HSA-5357956). Overall, analysis showed that the most relevant GO terms and biological pathways are associated with cytokine signalling and cascade inflammatory reactions. 

Moreover, results indicate that some of the candidate stroke modifiers in haemoglobinopathies are shared by stroke victims in the general population (e.g., *CCL2, F5, IL-6, SELP, TGFBR3, TNF*, and *VCAM1*). Large studies have been conducted to identify genes affecting stroke risk in the general population, resulting in the development of several biomarker panels that aim to risk stratify patients according to stroke type and to provide prognostic information for targeted interventions (biomarker panels are summarized in [98]). As many biomarkers are not disease-specific, diagnostic sensitivity and specificity is compromised [99,100,101]. The present work reveals the most prevalent biomarkers for stroke known to date in haemoglobinopathies. The stroke phenotype in IthaGenes comprises different stroke sub-types, such as large and small vessel types. Based on published reports of a variable genetic component across different types of stroke [60] and towards the development of a comprehensive account on stroke genetics, future work will focus on investigating genetic modifiers for each type of stroke separately. Knowledge on stroke biomarkers specific to haemoglobinopathies could serve as a guiding tool to assess future risk and to elucidate potential stroke pathways towards more effective personalised therapy.

## 4. Conclusions

Haemoglobinopathies are a heterogeneous group of Hb disorders characterised by diverse phenotypic manifestations. Despite considerable progress in accumulating knowledge on the genetic architecture of these phenotypes, the role of modulating genes on phenotypic expression is largely unclear. Deciphering gene–phenotype interactions is a crucial step in understanding disease pathology. The present work aims to highlight potential genes and molecular pathways that could explain the pathogenesis and complexity of haemoglobinopathy-specific phenotypes.

Using data from the IthaGenes database, a gene scoring algorithm (IthaScore) was developed to assist in evidence-based ranking of genetic modifiers for disease phenotypes. Gene scores were based on manual curation using a point system to collate and grade heterogeneous information and replication studies for each gene–phenotype relationship, with quantitative and qualitative evaluation of available evidence. IthaScore will be dynamically recalculated with emerging evidence for existing or new phenotype relationships and provides a measure of the volume and quality of evidence for such relationships. It does not provide any information about the size of the disease-modifying effect of any gene on the corresponding phenotype but can be a useful tool for gene ranking specific to haemoglobinopathies and their relevant phenotypes. This algorithm was validated in part by its ability to rank well-established genetic modifiers with high scores, such as the major QTLs (*BCL11A*, *HMIP* and *HBG2*) of Hb F production, one of the greatest markers and best-studied modulators of disease severity. 

To our knowledge, this is the first study integrating literature curation, gene ranking and functional enrichment analysis to evaluate candidate genetic modifiers for haemoglobinopathies. While we demonstrate the capacity of this approach to identify novel information, we urge caution when utilising the presented results due to potential limitations. Several of gene-phenotype findings are reported once without further replication, or they exhibit inconsistencies across studies due to differences in data collection and processing approaches. This is reflected by the low IthaScore calculated for the majority of gene-phenotype relationships. Moreover, most of published whole-genome scans do not always identify the true disease-modifying QTLs across large genomic regions, while findings may also come from studies with a small sample size and/or limited phenotyping, which are prone to noise [102,103]. Although IthaGenes harbours the largest collection of literature and continuously updated data on genetic modifiers for currently 59 phenotypes, additional genes may influence phenotypes that are not haemoglobinopathy-specific but are relatively common in the general population, such as stroke and osteoporosis.

In conclusion, the functional enrichment analysis for three phenotypes specific to β-thalassaemia and/or SCD provides preliminary proof that IthaGenes, as a comprehensive and scalable knowledgebase of genetic modifiers in haemoglobinopathies, together with dynamic gene scoring, can be used to unravel the molecular underpinnings of phenotypic diversity and identify new genes with plausible influence on haemoglobinopathy-specific phenotypes. Our findings add to current scientific knowledge and set the basis for future investigations. Research towards the discovery of phenotype-specific biomarkers will inform affected individuals about their health risk and allow them to thoughtfully consider their treatment options particularly with regards to stem cell transplantation and gene therapy, which offer the promise of complete cure albeit at a risk. Overall, the characterisation of candidate modifiers presents a novel and exciting opportunity to identify stratification biomarkers that help define treatment subgroups of patients in the frame of personalised medicine, as well as new diagnostic and therapeutic gene targets.

## Figures and Tables

**Figure 1 jcm-08-01927-f001:**
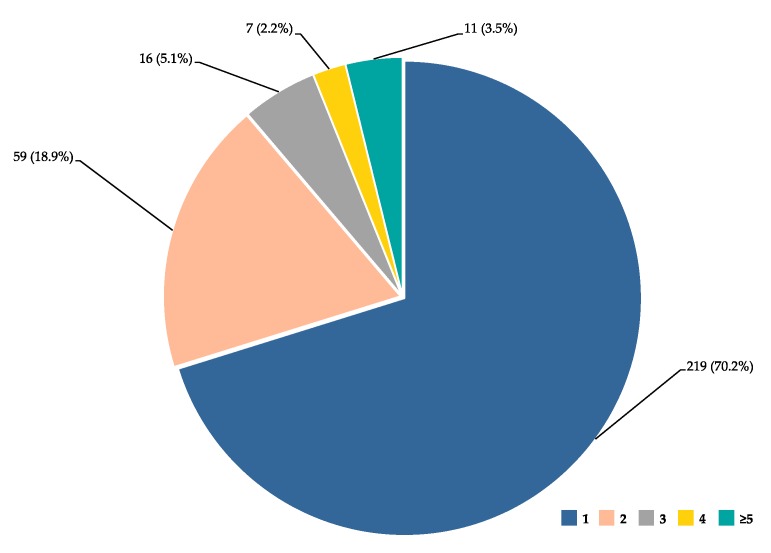
Gene distribution per phenotype annotations. Pie diagram shows the distribution of genes (*n* (%), for a total of 312 genes) based on the number of assigned phenotypic terms (1, 2, 3, 4, and ≥5).

**Figure 2 jcm-08-01927-f002:**
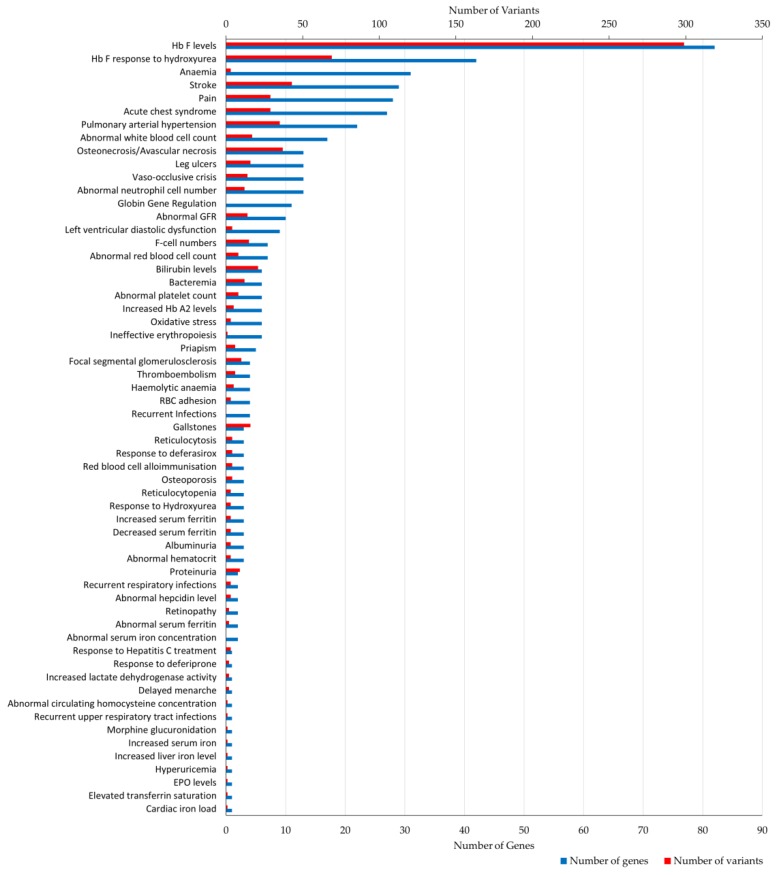
Number of gene and variant annotations per phenotypic term. Bar plot illustrates the number of genes (bottom x-axis, shown in blue) and variants (top x-axis, shown in red) assigned to each phenotypic term (total of 59) stored in IthaGenes. GFR, glomerular filtration rate; RBC, red blood cell; EPO, erythropoietin.

**Figure 3 jcm-08-01927-f003:**
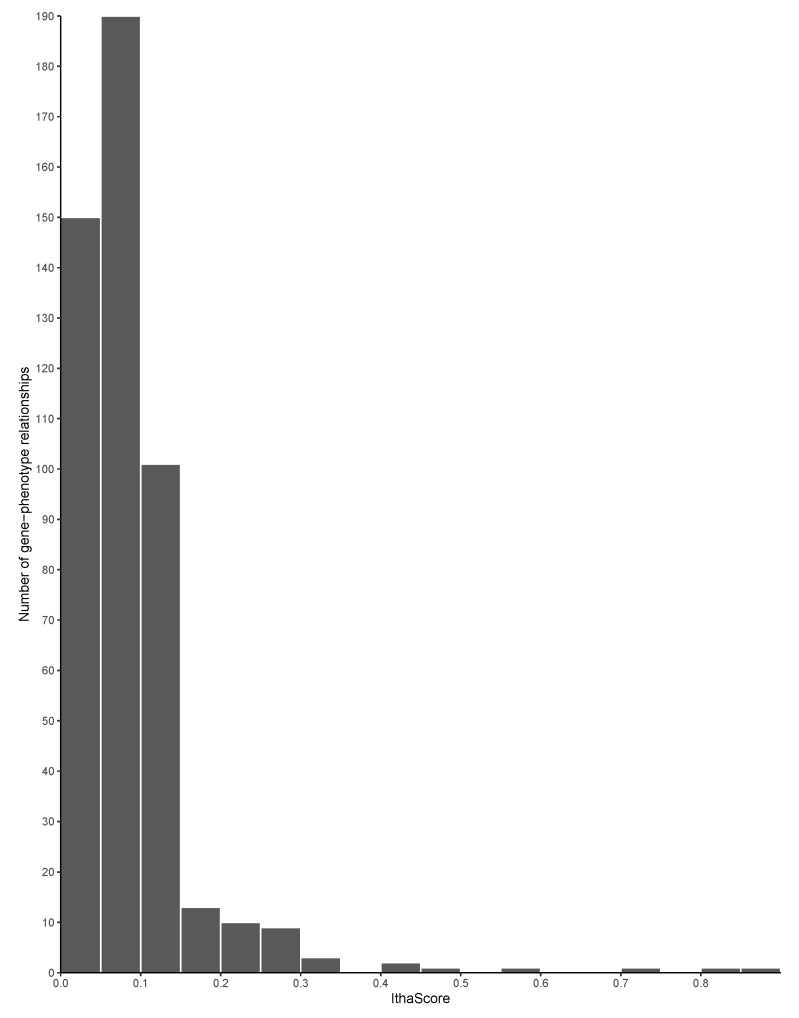
Distribution of IthaScore. Histogram shows the distribution of IthaScore in the range 0–1 for 483 gene-phenotype interactions.

**Figure 4 jcm-08-01927-f004:**
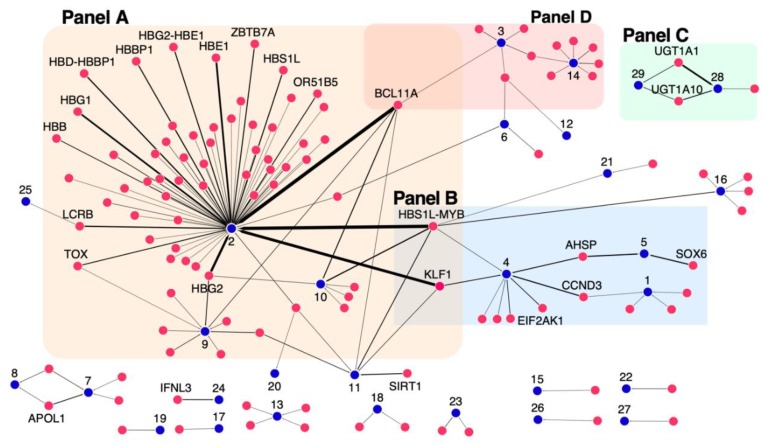
Network diagram of gene–phenotype interactions. The network depicts relationships between genes and phenotypes of haemoglobinopathies. Genes (red nodes) are connected to phenotypes (blue nodes) by edges. The thickness of the edges represents the corresponding IthaScore, where a stronger edge indicates a greater weight for the gene-phenotype relationship. Only gene–phenotype relationships with gene scores ≥0.1 are displayed on the network for better visualisation, while gene names are only shown for gene scores ≥0.2. The “Phenotype ID” shown in Table 2 is used to label phenotypic nodes, as follows: (1) Abnormal red blood cell count, (2) Hb F levels, (3) Stroke, (4) Anaemia, (5) Ineffective erythropoiesis, (6) Osteonecrosis/Avascular necrosis, (7) Focal segmental glomerulosclerosis, (8) Proteinuria, (9) Hb F response to hydroxyurea, (10) F-cell numbers, (11) Globin gene regulation, (12) Bacteremia, (13) Abnormal white blood cell count, (14) Acute chest syndrome, (15) Osteoporosis, (16) Abnormal platelet count, (17) Left ventricular diastolic dysfunction, (18) Pain, (19) Abnormal serum iron concentration, (20) Hyperuricemia, (21) Abnormal hematocrit, (22) Increased serum ferritin, (23) Vaso-occlusive crisis, (24) Response to Hepatitis C treatment, (25) Increased Hb A2 levels, (26) Erythropoietin (EPO) levels, (27) Haemolytic anaemia, (28) Bilirubin levels and (29) Gallstones.

**Figure 5 jcm-08-01927-f005:**
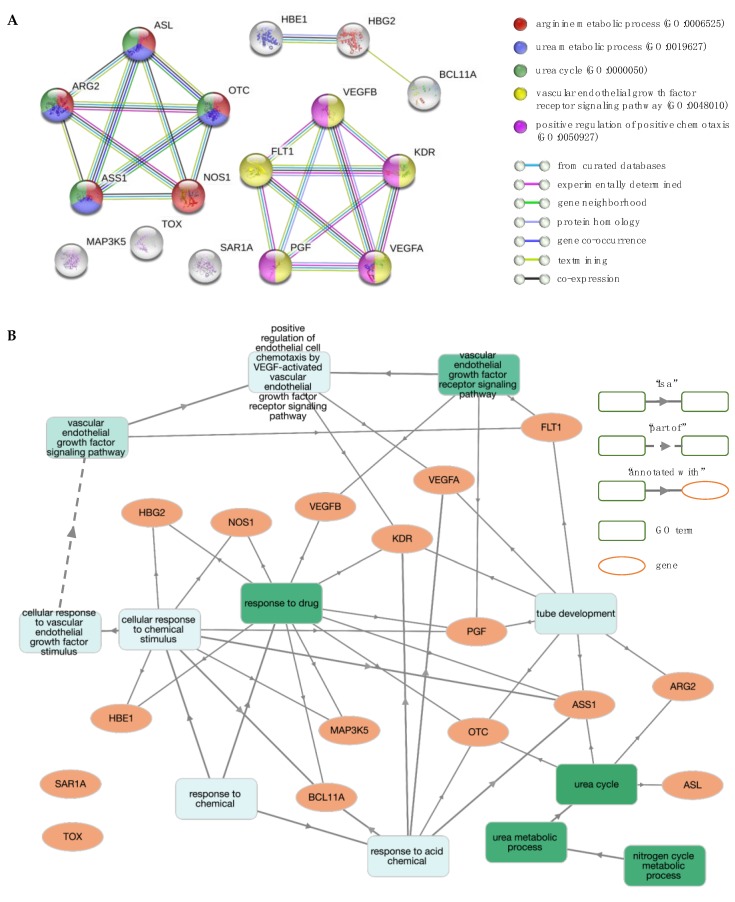
Network and enrichment analysis for Hb F levels and Hb F response to hydroxyurea (HU). (**A**) The protein–protein interaction (PPI) network contains 16 nodes (proteins; circles) connected by edges (protein–protein interactions). The most significant gene ontology (GO) biological process (BP) terms are shown. (**B**) GO BP enrichment analysis using GOnet (*q* value ≤0.01; *p* value ≤7.88 × 10^–6^). Green colour intensifies as the significance level of enrichment decreases.

**Figure 6 jcm-08-01927-f006:**
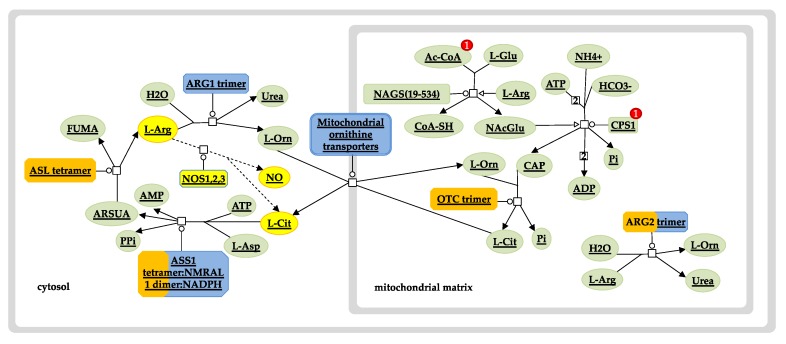
Urea cycle and nitric oxide pathway. Diagram depicts enzymes and intermediates of the urea cycle (solid lines) and the nitric oxide (NO) pathway (dashed lines, yellow nodes). Overrepresentation of the Hb F-related gene set used in the analysis is shown in orange. The size of the orange strip increases with the level of gene representation in the query set. The urea cycle pathway was exported from the Reactome pathway database and edited to include and highlight the role of NO shown in yellow. Ac-CoA, acetyl coenzyme A; AMP, adenosine monophosphate; ARG1, arginase 1; ARG2, arginase 2; ARSUA, argininosuccinate; ASL, arginosuccinate lyase; ASS1, arginosuccinate synthase; ATP, adenosine triphosphate; CAP, carbamoyl phosphate; CoA-SH, coenzyme A; CPS1, carbamoyl phosphate synthase 1; FUMA, fumarate; L-Arg, L-arginine; L-Asp, L-aspartate; L-Cit, L-citrulline; L-Glu, L-glutamine; L-Orn, L-ornithine; NAcGlu, N-acetylglutamic acid; NAGS, N-acetylglutamate synthase; NOS1,2,3, nitric oxide synthase 1 (neuronal, nNOS), 2 (inducible, iNOS), 3 (endothelial, eNOS); OTC, ornithine transcarbamylase; Pi, inorganic phosphate; PPi, inorganic pyrophosphate.

**Figure 7 jcm-08-01927-f007:**
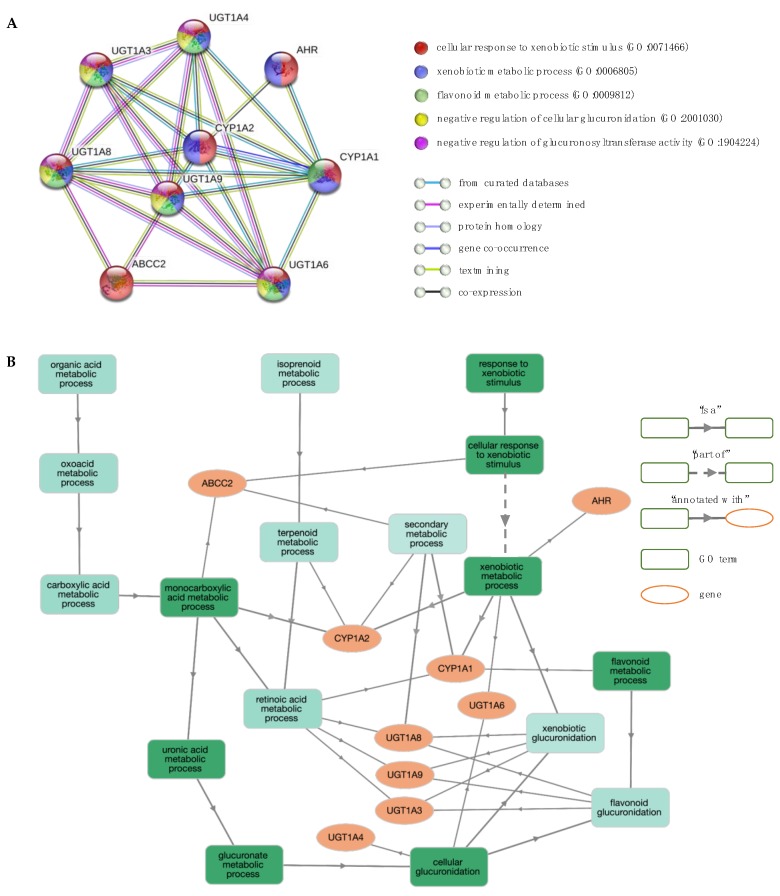
Network and enrichment analysis for response to iron chelators. (**A**) The PPI network contains nine nodes (proteins; circles) connected by edges (protein–protein interactions). The most significant GO BP terms are shown. (**B**) GO BP enrichment analysis using GOnet (*q* value ≤0.01; *p* value ≤5.3 × 10^–9^). Green colour intensifies as the significance level of enrichment decreases.

**Figure 8 jcm-08-01927-f008:**
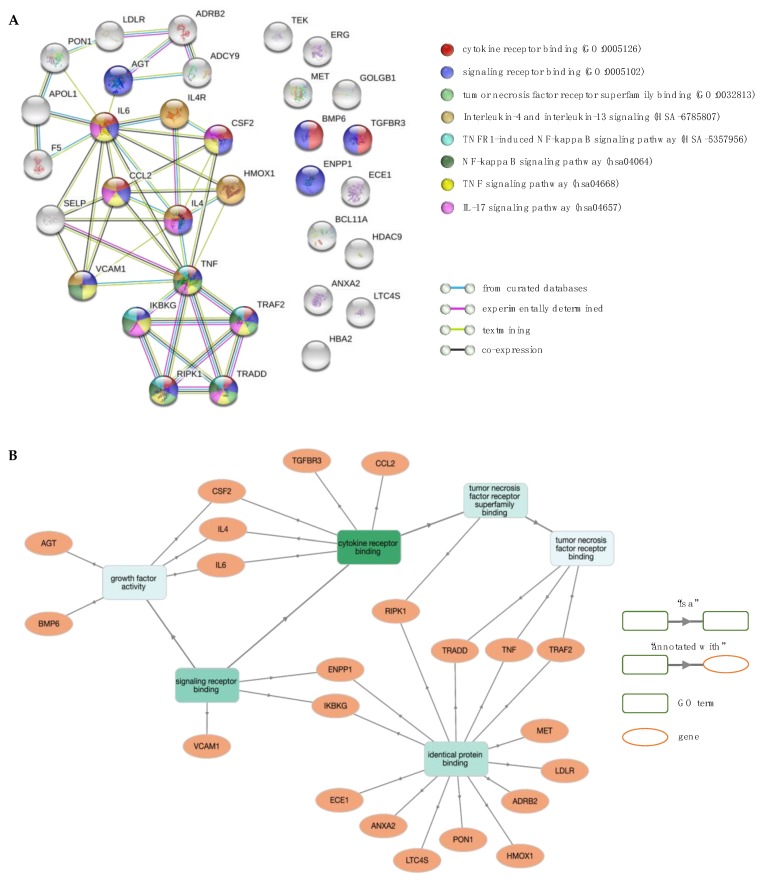
Network and enrichment analysis for stroke. (**A**) The PPI network contains 33 nodes (proteins; circles) connected by edges (protein–protein interactions; horizontal lines). Coloured nodes highlight proteins associated with significant molecular function (MF) terms and biological pathways. (**B**) GO MF enrichment analysis using GOnet (*q* value ≤0.01; *p* value ≤1.94 × 10^–5^). Green colour intensifies as the significance level of enrichment decreases.

**Table 1 jcm-08-01927-t001:** The point system used to score available evidence and to calculate the three individual scores (Association Score, Variant Score and Experimental Score) involved in the calculation of IthaScore. The point system was based on a similar approach described in References [40,46].

	**Evidence**	**Type**	**Description**	**Points**
**Association Score (AS)**	**Association study**	*p* value	<0.05	0.5
			<0.001	1
			<0.00001	1.5
	**Maximum Allowable Sum of Points for Association Score**			**8**
**Varian**t Score (VS)	**Genetic variants**	Number of variants	One point for each variant in every phenotype stored in IthaGenes.	1
	**Maximum Allowable Sum of Points for Variant Score**			**20**
**Experimental Score (ES)**	**Function**	Biochemical Function	Functions are shared between gene products involved in the same disease phenotype.	1
		Protein Interaction	Gene product interacts with proteins previously implicated in the disease phenotype. Gene defect disrupting protein interactions.	1
		Expression	Gene is expressed in tissues relevant to the disease phenotype. Altered gene expression in patients.	1
	**Functional Alteration**	Cells from affected individual	Function of gene product is altered in individuals/engineered cells with candidate mutations (altered expression levels, splicing or normal biochemical function).	1.5
		Engineered cells		1.5
	**Model Systems**	Animal model	Introduction of the variant or an engineered gene product carrying the variant in a non-human animal model/cell-culture model displays the disease phenotype.	2
		Cell culture model system		2
	**Rescue**	Rescue in non-human model organism	Addition of the wild-type gene product or specific knockdown of the variant allele can rescue the disease phenotype in a non-human model organism/cell-culture model/patient.	2
		Rescue in cell culture model		2
		Rescue in patients		2
	**Maximum Allowable Sum of Points for Experimental Score**			**6**

**Table 2 jcm-08-01927-t002:** The gene with the highest IthaScore for each phenotype. “Phenotype ID” column indicates the identifier assigned to each phenotype throughout this work.

Phenotype ID	Phenotypic Term	HPO ID	Gene/Intergenic Region	IthaScore
2	Hb F levels	HP:0011904	BCL11A	0.8750
28	Bilirubin levels	−	UGT1A1	0.4397
10	F-cell numbers	−	HBS1L-MYB	0.3169
5	Ineffective erythropoiesis	HP:0010972	AHSP, SOX6	0.3000
4	Anaemia	HP:0001903	CCND3	0.2938
11	Globin gene regulation	−	SIRT1	0.2500
9	Hb F response to hydroxyurea	−	HBG2	0.2188
7	Focal segmental glomerulosclerosis	HP:0000097	APOL1	0.2175
24	Response to Hepatitis C treatment	−	IFNL3	0.2175
16	Abnormal platelet count	HP:0011873	HBS1L-MYB	0.1997
29	Gallstones	HP:0001081	UGT1A1	0.1663
14	Acute chest syndrome	−	EDN1	0.1450
23	Vaso-occlusive crisis	−	HMOX1	0.1413
6	Osteonecrosis/Avascular necrosis	HP:0010885	KL	0.1413
3	Stroke	HP:0001297	ENPP1	0.1350
22	Increased serum ferritin	HP:0003281	HFE	0.1350
8	Proteinuria	HP:0000093	MYH9	0.1325
19	Abnormal serum iron concentration	HP:0040130	GDF15	0.1250
18	Pain	HP:0012531	GCH1	0.1184
17	Left ventricular diastolic dysfunction	HP:0025168	FUCA2	0.1100
1	Abnormal red blood cell count	HP:0020058	ABO, CCND3, PRKCE, PARP11-CCND2	0.1038
13	Abnormal white blood cell count	HP:0011893	CDK6, LY6G5C, PNPLA3, PSMD3-CSF3	0.1038
20	Hyperuricemia	HP:0002149	HBG1-HBG2	0.1038
21	Abnormal hematocrit	HP:0031850	HBS1L-MYB, PDGFRA-KIT	0.1038
25	Increased Hb A2 levels	HP:0045048	LCRB	0.1038
27	Haemolytic anaemia	HP:0001878	NPRL3	0.1038
26	EPO levels	−	MAP2K6	0.1038
15	Osteoporosis	HP:0000939	COL1A1	0.1038
12	Bacteremia	HP:0031864	BMP6	0.1025
30	Oxidative stress	HP:0025464	FOXO3	0.1000
31	Albuminuria	HP:0012592	APOL1	0.0959
32	Pulmonary arterial hypertension	HP:0002092	NEDD4L	0.0825
33	RBC adhesion	−	ADCY6	0.0825
34	Delayed menarche	HP:0012569	NOS3	0.0825
35	Red blood cell alloimmunisation	−	CD81	0.0825
36	Reticulocytosis	HP:0001923	NPRL3	0.0803
37	Abnormal neutrophil cell number	HP:0011991	NES	0.0803
38	Abnormal GFR	HP:0012212	APOL1	0.0747
39	Leg ulcers	−	SMAD7	0.0725
40	Increased serum iron	HP:0003452	HFE	0.0725
41	Cardiac iron load	−	GSTM1	0.0725
42	Thromboembolism	HP:0001907	PROC	0.0613
43	Response to Hydroxyurea	−	CD36	0.0600
44	Priapism	HP:0200023	AQP1, ITGAV, TGFBR3	0.0569
45	Reticulocytopenia	HP:0001896	BCL11A	0.0569
46	Recurrent respiratory infections	HP:0002205	LGALS3	0.0513
47	Increased lactate dehydrogenase activity	HP:0025435	NOS3	0.0513
48	Response to deferiprone	−	UGT1A6	0.0513
49	Abnormal hepcidin level	HP:0031875	TMPRSS6	0.0434
50	Abnormal serum ferritin	HP:0040133	GSTM1	0.0413
51	Elevated transferrin saturation	HP:0012463	HFE	0.0413
52	Decreased serum ferritin	HP:0012343	TF, TFR2, TNF	0.0413
53	Abnormal circulating homocysteine concentration	HP:0010919	MTHFR	0.0413
54	Morphine glucuronidation	−	UGT2B7	0.0413
55	Increased liver iron level	HP:0012465	HAMP	0.0334
56	Response to deferasirox	−	CYP1A2	0.0434
57	Retinopathy	HP:0000488	IL6, NOS3	0.0413
58	Recurrent upper respiratory tract infections	HP:0002788	NOS3	0.0413
59	Recurrent Infections	HP:0002719	CCL5, MPO, TLR2	0.0313

Abbreviations: EPO, erythropoietin; GFR, glomerular filtration rate; Hb, haemoglobin; HPO, Human Phenotype Ontology; RBC, red blood cell.

**Table 3 jcm-08-01927-t003:** Top 10 gene-phenotype interactions with the highest IthaScore. “Phenotype ID” column represents the identifier assigned to each phenotype throughout this work.

Phenotype ID	Phenotypic Term	HPO ID	Gene/Intergenic Region	IthaScore
2	Hb F levels	HP:0011904	BCL11A	0.875
2	Hb F levels	HP:0011904	HBS1L-MYB	0.825
2	Hb F levels	HP:0011904	KLF1	0.711
2	Hb F levels	HP:0011904	HBG2	0.600
2	Hb F levels	HP:0011904	HBE1	0.462
28	Bilirubin levels	−	UGT1A1	0.440
2	Hb F levels	HP:0011904	HBG1	0.435
2	Hb F levels	HP:0011904	HBD-HBBP1	0.330
10	F-cell numbers	−	HBS1L-MYB	0.317
2	Hb F levels	HP:0011904	LCRB	0.312

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
