# Peer review of "Genetic Modifiers at the Crossroads of Personalised Medicine for Haemoglobinopathies"

_jcm, 2019, doi:10.3390/jcm8111927_

Round 1
Reviewer 1 Report
line 63-64- While elevated Hb F levels have no clinical benefit to healthy adults, they have been demonstrated to ameliorate disease severity- please clarify that by healthy adults you are referring to those without a haemoglobinopathy
line 71/72
Facilitated by the advent of technology, recent studies [19–21] have identified variants associated with laboratory and clinical markers of disease severity, aiming at defining potential biomarkers that would help risk-stratify patients to direct care, assist with early screening and diagnosis of symptoms, adjust dosing regimens for safe and effective drug thera- this point need further development- what are the markers?
Line 143-148
The volume of available evidence for each gene-phenotype relationship in the dataset is represented quantitatively with three different scores, namely Association Score, Variant Score and Experimental Score-a)case-level studies and association studies reporting statistically significant associations with a P value of < 0.05, scored with 0.5 point, (b) association studies with at least one variant with a P value of < 0.001, scored with 1 point, and (c) association studies with at least one variant with a P value of < 10-5, scored with 1.5 points. To avoid possible bias from multiple case- level studies, a maximum of four such studies (i.e. a total of 2 points) were considered for each gene- phenotype relationship. if one study can have up to 1.5 how will 4 studies have a maximum score of 2 only, this is different from the table which says 8. If so how do you achieve 8 from 1.5 max from 4 studies?
330 / 33`Genes (red nodes) are connected to phenotypes (blue nodes) by edges. Change nodes to dots? This linkage appears to be an important step that requires more detailed explanation. It would help readers if the blue dots are also named.
Line 469/470
The clinical and commercial advantages of iron chelators are huge, fueling pharmacogenetic research to identify novel drug-response proteins.- this is not relevant
Line 481
Stroke is one of the most devastating complications of SCD affecting up to 11% of patients and having over 60% of recurrence after an initial event without therapeutic intervention [84,85]—11% is the risk at 18 years for HbSS/Sβo not all SCD and not a life time risk either
There are too many undefined terminologies
GO, STRING, PPI, etc this makes harder for non geneticists to grasp
generally I would suggest that this article is reviewed by someone with a good expertise in genetic studies. it is an important concept that is relevant to the generalists. I was excited to see an attempt to correlate genotype with phenotype and help unravel the underlying factors accounting for the variability especially in sickle cell disease. a platform like this will help in targeting research objectives to delineate risk factors and plan therapeutic targets.
Author Response
We would like to thanks the reviewer for the useful comments. Please see the attachment for our reply.

Reviewer 2 Report
The authors of the paper showed an excellent model of interaction between biological and clinical aspects of haemoglobinopathies. For monogenic disorders with great phenotypic variability as haemoglobinopathies, the complementarity of molecular data with clinical features is essential for patient management.
I have some minor comments:
Line 20: Gene Ontology (GO) and so for all those that are used for the first time in the text
Lines 74-75: Please mention other publications
Fig.1 Gene distribution per phenotype annotations: It is advisable to use different colors for pie diagram
Fig.3 Distribution of IthaScore: It is advisable to use different colors for histogram
Fig. 5B and Fig. 7B: The continuous and dashed arrows cannot be distinguished
Line 485: Please mention other publications, for example: Martella M, Quaglia N, Frigo AC, Basso G, Colombatti R, Sainati L. Association between a combination of single nucleotide polymorphisms and large vessel cerebral vasculopathy in African children with sickle cell disease. Blood Cells Mol Dis. 2016 Oct;61:1-3.
Lines 522-524: Please mention other publications
Author Response
We would like to thank the reviewer for the useful comments. Please see the attachment for our reply.
